# Nest Your Adaptive Algorithm for Parameter-Agnostic Nonconvex Minimax Optimization

**Junchi Yang**\*
Department of Computer Science
ETH Zurich, Switzerland
junchi.yang@inf.ethz.ch

**Xiang Li**\*
Department of Computer Science
ETH Zurich, Switzerland
xiang.li@inf.ethz.ch

**Niao He**
Department of Computer Science
ETH Zurich, Switzerland
niao.he@inf.ethz.ch

## Abstract

Adaptive algorithms like AdaGrad and AMSGrad are successful in nonconvex optimization owing to their *parameter-agnostic* ability – requiring no a priori knowledge about problem-specific parameters nor tuning of learning rates. However, when it comes to nonconvex minimax optimization, direct extensions of such adaptive optimizers without proper *time-scale separation* may fail to work in practice. We provide such an example proving that the simple combination of Gradient Descent Ascent (GDA) with adaptive stepsizes can diverge if the primal-dual stepsize ratio is not carefully chosen; hence, a fortiori, such adaptive extensions are not parameter-agnostic. To address the issue, we formally introduce a Nested Adaptive framework, NeAda for short, that carries an inner loop for adaptively maximizing the dual variable with controllable stopping criteria and an outer loop for adaptively minimizing the primal variable. Such mechanism can be equipped with off-the-shelf adaptive optimizers and automatically balance the progress in the primal and dual variables. Theoretically, for nonconvex-strongly-concave minimax problems, we show that NeAda with AdaGrad stepsizes can achieve the near-optimal $\widetilde{O}(\epsilon^{-2})$ and $\widetilde{O}(\epsilon^{-4})$ gradient complexities respectively in the deterministic and stochastic settings, *without* prior information on the problem's smoothness and strong concavity parameters. To the best of our knowledge, this is the first algorithm that simultaneously achieves near-optimal convergence rates and parameter-agnostic adaptation in the nonconvex minimax setting. Numerically, we further illustrate the robustness of the NeAda family with experiments on simple test functions and a real-world application.

## 1 Introduction

Adaptive gradient methods, whose stepsizes and search directions are adjusted based on past gradients, have received phenomenal popularity and are proven successful in a variety of large-scale machine learning applications. Prominent examples include AdaGrad [17], RMSProp [31], AdaDelta [84], Adam [41], and AMSGrad [69], just to name a few. Their empirical success is especially pronounced for nonconvex optimization such as training deep neural networks. Besides improved performance, being *parameter-agnostic* is another important trait of adaptive methods. Unlike (stochastic) gradient descent, adaptive methods often do not require a priori knowledge about problem-specific parameters

---

\*Equal contribution.

36th Conference on Neural Information Processing Systems (NeurIPS 2022).

(such as Lipschitz constants, smoothness, etc.).[2] On the theoretical front, some adaptive methods can achieve nearly the same convergence guarantees as (stochastic) gradient descent [17, 79, 69].

Recently, adaptive methods have sprung up for minimax optimization:

$$\min_{x \in \mathbb{R}^d} \max_{y \in \mathcal{Y}} f(x, y) \triangleq \mathbb{E}[F(x, y; \xi)], \tag{1}$$

where $f$ is $l$-Lipschitz smooth jointly in $x$ and $y$, $\mathcal{Y}$ is closed and convex, and $\xi$ is a random vector. Such problems have found numerous applications in generative adversarial networks (GANs) [25, 4], Wasserstein GANs [4], generative adversarial imitation learning [32], reinforcement learning [14, 60], adversarial training [74], domain-adversarial training of neural networks [21] , etc.

A common practice is to simply combine adaptive stepsizes with popular minimax optimization algorithms such as Gradient Descent Ascent (GDA), extragradient method (EG) and the like; see e.g., [23, 27, 24]. It is worth noting that these methods are reported successful in some applications yet at other times can suffer from training instability. In recent years, theoretical behaviors of such adaptive methods are extensively studied for convex-concave minimax optimization; see e.g., [6, 3, 1, 18, 72, 22, 58, 15]. However, for minimax optimization in the important nonconvex regime, little theory related to adaptive methods is known.

Unlike the convex-concave setting, a key challenge for nonconvex minimax optimization lies in the necessity of a *problem-specific time-scale separation* of the learning rates between the min-player and max-player when GDA or EG methods are applied, as proven in [82, 50, 70, 8]. This makes the design of adaptive methods fundamentally different from and more challenging than nonconvex minimization. Several recent attempts [28, 33, 34] studied adaptive methods for nonconvex-strongly-concave minimax problems; yet, they all require explicit knowledge of the problems' smoothness and strong concavity parameters to maintain a stepsize ratio proportional to the condition number. Such a requirement evidently undermines the parameter-agnostic trait of adaptive methods. This then raises a couple of interesting questions: (1) *Without a problem-dependent stepsize ratio, does simple combination of GDA and adaptive stepsizes still converge?* (2) *Can we design an adaptive algorithm for nonconvex minimax optimization that is truly parameter-agnostic and provably convergent?*

In this paper, we address these questions and make the following key contributions:

- We investigate two generic frameworks for adaptive minimax optimization: one is a simple (non-nested) adaptive framework, which performs one step of update of $x$ and $y$ simultaneously with adaptive gradients; the other is Nested Adaptive (NeAda) framework, which performs multiple updates of $y$ after one update of $x$, each with adaptive gradients. Both frameworks allow flexible choices of adaptive mechanisms such as Adam, AMSGrad and AdaGrad. We provide an example proving that the simple adaptive framework can fail to converge without setting an appropriate stepsize ratio; this applies to any of the adaptive mechanisms mentioned above, even in the noiseless setting. In contrast, the NeAda framework is less sensitive to the stepsize ratio, as numerically illustrated in Figure 1.

- We provide the convergence analysis for a representative of NeAda that uses AdaGrad stepsizes for $x$ and a convergent adaptive optimizer for $y$, in terms of nonconvex-strongly-concave minimax problems. Notably, the convergence of this general scheme does not require to know any problem parameters and does not assume the bounded gradients. We demonstrate that NeAda is able to achieve $\widetilde{O}(\epsilon^{-2})$ oracle complexity for the deterministic setting and $\widetilde{O}(\epsilon^{-4})$ for the stochastic setting to converge to $\epsilon$-stationary point, matching best known bounds. To the best of our knowledge, this seems to be the first adaptive framework for nonconvex minimax optimization that is provably convergent and parameter-agnostic.

- We further make two complementary contributions, which can be of independent interest. First, we propose a general AdaGrad-type stepsize for strongly-convex problems without knowing the strong convexity parameters, and derive a convergence rate comparable to SGD. It can serve as a subroutine for NeAda. Second, we provide a high probability convergence result for the primal variable of NeAda under a subGaussian assumption.

---

[2]For distinction, we use "parameter-agnostic" to describe algorithms that do not ask for problem-specific parameters in setting their stepsizes or hyperparameters; we refer to "adaptive algorithms" as methods whose stepsizes are based on the previously observed gradients.

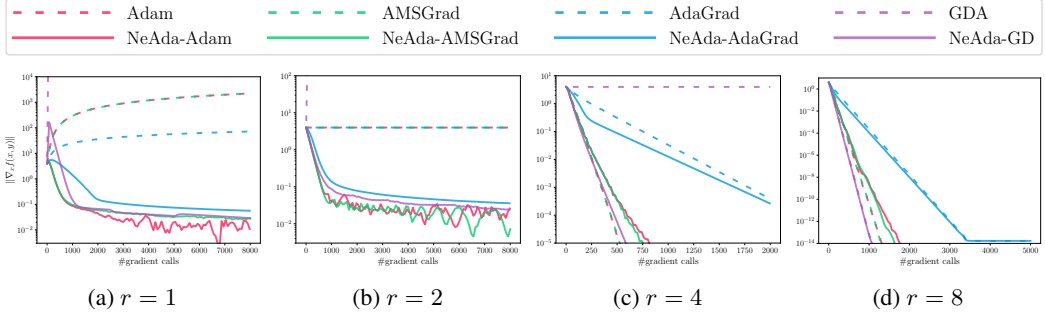

| (a) $r = 1$ | (b) $r = 2$ | (c) $r = 4$ | (d) $r = 8$ |

Figure 1: Comparison between the two families of non-nested and nested adaptive methods on function $f(x, y) = -\frac{1}{2}y^2 + 2xy - 2x^2$ with deterministic gradient oracles. $r = \eta^y / \eta^x$ is a pre-fixed learning rate ratio.

- Finally, we numerically validate the robustness of the NeAda framework on several test functions compared to the non-nested adaptive framework, and demonstrate the effectiveness of the NeAda framework on distributionally robust optimization task with a real dataset.

## 1.1 Related work

**Adaptive algorithms.** Duchi et al. [17] and Streeter and McMahan [73] introduce AdaGrad for convex online learning and achieve $O(\sqrt{T})$ regrets. Li and Orabona [48] and Ward et al. [79] show an $\widetilde{O}(\epsilon^{-4})$ complexity for AdaGrad in the nonconvex stochastic optimization. There are an extensive number of works on AdaGrad-type methods; to list a few, [59, 45, 2, 39, 65]. Another family of algorithms uses more aggressive stepsizes of exponential moving average of the past gradients, such as Adam [41] and RMSProp [31]. Reddi et al. [69] point out the non-convergence of Adam and provide a remedy with non-increasing stepsizes. There is a surge in the study of Adam-type algorithms due to their popularity in the deep neural network training [83, 11, 51]. Some work provides the convergence results for adaptive methods in the strongly-convex optimization [78, 44, 62]. Line search and stochastic line search are another effective strategy that can detect the objective's curvature and have received much attention [75, 77, 76]. Notably, many adaptive algorithms are parameter-agnostic [17, 69, 79].

**Nonconvex minimax optimization.** Stationary convergence of GDA in NC-SC setting was first provided by Lin et al. [50], showing $O(\epsilon^{-2})$ oracle complexity and $O(\epsilon^{-4})$ sample complexity with minibatch. Recently, Chen et al. [10] and Yang et al. [82] achieve this sample complexity in the stochastic setting without minibatch. GDmax is a double loop algorithm that maximizes the dual variable to a certain accuracy. It achieves nearly the same complexity as GDA [64]. Sebbouh et al. [70] recently discuss the relation between the two-time-scale and number of inner steps for GDmax. Very recently, Li et al. [47] show that time-scale separation is necessary for GDA to converge to Stackelberg equilibrium. Besides NC-SC setting, some work provides convergent algorithms when the objective is (non-strongly) concave about the dual variable[85, 55, 81]. Nonconvex-nonconcave regime is only explored under some special structure [53, 15], such as Polyak-Łojasiewicz (PL) condition [19]. All algorithms mentioned above require prior knowledge about problem parameters, such as smoothness modulus, strong concavity modulus, and noise variance.

**Adaptive algorithms in minimax optimization.** There exist many adaptive and parameter-agnostic methods designed for convex-concave minimax optimization as a special case of monotone variational inequality [6, 3, 1, 18, 72, 22, 58, 15]. Most of them combine extragradient method, mirror prox [63] or the like, with AdaGrad mechanism. Liu et al. [52] and Dou and Li [16] relax convexity-concavity assumption to the regime where Minty variational inequality (MVI) has a solution. In these settings, time-scale separation of learning rates is not required even for non-adaptive algorithms. For nonconvex-strongly-concave problems, Huang and Huang [33], Huang et al. [34], Guo et al. [28] propose adaptive methods, which set the learning rates based on knowledge about smoothness and strong-concavity modulus and the bounds for adaptive stepsizes.

## 2 Non-nested and nested adaptive methods

In this section, we investigate two generic frameworks that can incorporate most existing adaptive methods into minimax optimization. We remark that many variants encapsulated in these two families are already widely used in practice, such as training of GAN [24], distributionally robust optimization [71], etc. These two frameworks, coined as non-nested and nested adaptive methods, can be viewed as adaptive counterparts of GDA and GDmax. We aim to illustrate the difference between these two adaptive families, even though GDA and GDmax are often considered "twins".

**Non-nested adaptive methods.** In Algorithm 1, non-nested methods update the primal and dual variables in a symmetric way. Weighted gradients $m_t^x$ and $m_t^y$ are the moving average of the past stochastic gradients with the momentum parameters $\beta^x$ and $\beta^y$. The effective stepsizes of $x$ and $y$ are $\eta^x/\sqrt{v_t^x}$ and $\eta^y/\sqrt{v_t^y}$, where the division is taken coordinate-wise. We refer to $\eta^x$ and $\eta^y$ as learning rates, and $v_t^x, v_t^y$ are some average of squared-past gradients through function $\psi$. Many popular choices of adaptive stepsizes are captured in this framework, see also [69]:

$$\text{(GDA)} \quad \beta = 0; \ \psi\left(v_0, \{g_i^2\}_{i=0}^t\right) = 1, \quad \text{(AdaGrad)} \quad \beta = 0; \ \psi\left(v_0, \{g_i^2\}_{i=0}^t\right) = v_0 + \sum_{i=0}^t g_i^2,$$

$$\text{(Adam)} \quad \psi\left(v_0, \{g_i^2\}_{i=0}^t\right) = \gamma^{t+1}v_0 + (1-\gamma)\sum_{i=0}^t \gamma^{t-i}g_i^2,$$

$$\text{(AMSGrad)} \quad \psi\left(v_0, \{g_i^2\}_{i=0}^t\right) = \max_{m=0,\dots,t} \gamma^{m+1}v_0 + (1-\gamma)\sum_{i=0}^m \gamma^{m-i}g_i^2.$$

| **Algorithm 1** Non-nested Adaptive Method | **Algorithm 2** Nested Adaptive (NeAda) Method |
|---|---|
| 1: Input: $x_0$ and $y_0$ | 1: Input: $x_0$ and $y_0^0$ |
| 2: **for** $t = 0, 1, 2, \dots$ **do** | 2: **for** $t = 0, 1, 2, \dots$ **do** |
| 3:    sample $\xi_t$ and let 
    $g_t^x = \nabla_x F(x_t, y_t; \xi_t)$ and 
    $g_t^y = \nabla_y F(x_t, y_t; \xi_t)$ | 3:    **for** $k = 0, 1, 2, \dots$ until a stopping criterion is 
    satisfied **do** |
| 4:    // update the first moment 
    $m_{t+1}^x = \beta^x m_t^x + (1-\beta^x)g_t^x$ and 
    $m_{t+1}^y = \beta^y m_t^y + (1-\beta^y)g_t^y$ | 4:      sample $\hat{\xi}_t^k$ and $g_{t,k}^y = \nabla_y F(x_t, y_t^k; \hat{\xi}_t^k)$ 
 5:      $m_{t,k+1}^y = \beta^y m_{t,k}^y + (1-\beta^y)g_{t,k}^y$ |
| 5:    // update the second moment 
    $v_{t+1}^x = \psi\left(v_0^x, \{(g_i^x)^2\}_{i=0}^t\right)$ and 
    $v_{t+1}^y = \psi\left(v_0^y, \{(g_i^y)^2\}_{i=0}^t\right)$ | 6:      $v_{t,k+1}^y = \psi^y\left(v_{t,0}^y, \{(g_{t,i}^y)^2\}_{i=0}^k\right)$ 
 7:      $y_t^{k+1} = y_t^k + \frac{\eta^y}{\sqrt{v_{t,k+1}^y}}m_{t,k+1}^y$ |
| 6:    // update variables 
    $x_{t+1} = x_t - \frac{\eta^x}{\sqrt{v_{t+1}^x}}m_{t+1}^x$ and 
    $y_{t+1} = y_t + \frac{\eta^y}{\sqrt{v_{t+1}^y}}m_{t+1}^y$ | 8:    **end for** 
 9:    $v_{t+1,0}^y = v_{t,k+1}^y$ and $m_{t+1,0}^y = m_{t,k+1}^y$ 
 10:   sample $\xi_t$ and $g_t^x = \nabla_x F(x_t, y_t^{k+1}; \xi_t)$ 
 11:   $m_{t+1}^x = \beta^x m_t^x + (1-\beta^x)g_t^x$ 
 12:   $v_{t+1}^x = \psi^x\left(v_0^x, \{(g_i^x)^2\}_{i=0}^t\right)$ 
 13:   $x_{t+1} = x_t - \frac{\eta^x}{\sqrt{v_{t+1}^x}}m_{t+1}^x$ |
| 7: **end for** | 14: **end for** |

**Nested adaptive (NeAda) methods.** NeAda, presented in Algorithm 2, has a nesting inner loop to maximize $y$ until some stopping criterion is reached (see details in Section 3). Instead of using a fixed number of inner iterations or a fixed target accuracy as in GDmax [50, 64], NeAda gradually increases the accuracy of the inner loop as the outer loop proceeds to make it fully adaptive.

We refer to the ratio between two learning rates, i.e. $\eta^y/\eta^x$, as the two-time-scale. The current analysis of GDA in nonconvex-strongly-concave setting requires two-time-scale to be proportional with the condition number $\kappa = l/\mu$, where $l$ and $\mu$ are Lipschitz smoothness and strongly-concavity modulus [50, 82]. We provide an example showing that the problem-dependent two-time-scale is *necessary* for GDA and most non-nested methods even in the deterministic setting.

**Lemma 2.1.** *Consider the function $f(x, y) = -\frac{1}{2}y^2 + Lxy - \frac{L^2}{2}x^2$ in the deterministic setting. Let $r\eta^x = \eta^y$. (1) GDA will not converge to the stationary point when $r \leq L^2$:*

$$\nabla_x f(x_T, y_T) = \nabla_x f(x_0, y_0) \prod_{t=0}^{T-1} \left[1 + \eta^x(L^2 - r)\right].$$

*(2) Assume the averaging function $\psi^x$ and $\psi^y$ are the same, and satisfy that for any $\tau$, if $v_t^x = \tau v_t^y$ and $(g_t^x)^2 = \tau (g_t^y)^2$ then $v_{t+1}^x = \tau v_{t+1}^y$. With $\beta^x = \beta^y$, $v_0^x = v_0^y = 0$ and $m_0^x = m_0^y = 0$ (which are commonly used in practice), non-nested adaptive method will not converge when $r \leq L$:*

$$\nabla_x f(x_T, y_T) \geq \nabla_x f(x_0, y_0) \prod_{t=0}^{T-1} \left[ 1 + \frac{L \eta^x}{\sqrt{v_t^x}} (1 - \beta^x)(L - r) \right].$$

*When $r = L$, $\nabla_x f(x_t, y_t) = \nabla_x f(x_0, y_0)$ for all $t$.*

**Remark 1.** *Most popular adaptive stepsizes we mentioned before, such as Adam, AMSGrad and AdaGrad, have averaging functions satisfying the assumption in the lemma. Any point on the line $y = Lx$ is a stationary point for the above function, and the distance from a point to this line is proportional to its gradient norm, so the divergence in gradient norm will also implies that of iterates. In the proof, we will also show that the averaged or best iterate will still diverge under the same condition. The lemma implies that for any given time-scale $r$, there exists a problem for which the non-nested algorithm does not converge to the stationary point, so they are not parameter-agnostic.*

We compare non-nested and nested methods combined with different stepsizes schemes: Adam, AMSGrad, AdaGrad and fixed stepsize, on the function: $-\frac{1}{2} y^2 + 2xy - 2x^2$. In the experiments of this section, we halt the inner loop when the (stochastic) gradient about $y$ is smaller than $1/t$ or the number iteration is greater than $t$. We observe from Figure 1 that the thresholds for the non-convergence of non-nested methods ($r = 2$ for adaptive methods and $r = 4$ for GDA) are exactly as predicted by the lemma. Although the adaptive methods admit a smaller two-time-scale threshold than GDA in this example, it is not a universal phenomenon from our experiments in Section 4. Interestingly, nested adaptive methods are robust to different two-time-scales and always have the trend to converge to the stationary point.

## 3  Convergence Analysis of NeAda-AdaGrad

In this section, we reveal the secret behind the robust performance of NeAda by providing the convergence guarantee for a representative member in the family. For sake of simplicity and clarity, we mainly focus on NeAda with AdaGrad. Adam-type mechanism can suffer from non-convergence already for nonconvex minimization despite its good performance in practice. Our result also sheds light on the analysis of other more sophisticated members such as AMSGrad in the family.

**NeAda-AdaGrad:**  Presented in Algorithm 3, NeAda-AdaGrad adopts the scalar AdaGrad scheme [73] for the $x$-update in the outer loop and uses mini-batch in the stochastic setting. For the inner loop for maximizing $y$, we run some adaptive algorithm for maximizing $y$ until some easily checkable stopping criterion is satisfied. We suggest two criteria here: at $t$-th outer loop: (I) the squard gradient mapping norm about $y$ is smaller than $1/(t+1)$ in the deterministic setting, (II) the number of inner loop iterations reaches $t + 1$ in the stochastic setting.

---

**Algorithm 3** NeAda-AdaGrad

1: Input: $(x_0, y_{-1})$, $v_0 > 0$, $\eta > 0$.
2: **for** $t = 0, 1, 2, ..., T - 1$ **do**
3:  from $y_{t-1}$ run an adaptive algorithm $\mathcal{A}$ for maximizing $f(x_t, \cdot)$ to obtain $y_t$
   (a) stopping criterion I (deterministic): stop when $\|y_t - \text{Proj}_{\mathcal{Y}}(y_t + \nabla_y f(x_t, y_t))\|^2 \leq \frac{1}{t+1}$
   (b) stopping criterion II (stochastic): stop after $t + 1$ inner loop iterations.
4:  $v_{t+1} = v_t + \left\| \frac{1}{M} \sum_{i=1}^{M} \nabla_x F(x_t, y_t; \xi_t^i) \right\|^2$ where $\{\xi_t^i\}_{i=1}^{M}$ are i.i.d samples
5:  $x_{t+1} = x_t - \frac{\eta}{\sqrt{v_{t+1}}} \left( \frac{1}{M} \sum_{i=1}^{M} \nabla_x F(x_t, y_t; \xi_t^i) \right)$
6: **end for**

---

For the purpose of theoretical analysis, we mainly focus on the minimax problem of the form (1) under the nonconvex-strongly-concave (NC-SC) setting[3], formally stated in the following assumptions.

---

[3]Note that for other nonconvex minimax optimization beyond the NC-SC setting, even the convergence of non-adaptive gradient methods has not been fully understood.

**Assumption 3.1** (Lipschitz smoothness). *There exists a positive constant $l > 0$ such that*

$$\max\left\{ \|\nabla_x f\left(x_1, y_1\right) - \nabla_x f\left(x_2, y_2\right)\|, \|\nabla_y f\left(x_1, y_1\right) - \nabla_y f\left(x_2, y_2\right)\| \right\} \le l[\|x_1 - x_2\| + \|y_1 - y_2\|],$$

*holds for all $x_1, x_2 \in \mathbb{R}^d, y_1, y_2 \in \mathcal{Y}$.*

**Assumption 3.2** (Strong-concavity in $y$). *There exists $\mu > 0$ such that: $f(x, y_1) \ge f(x, y_2) + \langle \nabla_y f(x, y_1), y_1 - y_2 \rangle + \frac{\mu}{2}\|y_1 - y_2\|^2, \forall x \in \mathbb{R}^d, y_1, y_2 \in \mathcal{Y}$.*

For simplicity of notation, define $\kappa = l/\mu$ as the condition number, $\Phi(x) = \max_{y \in \mathcal{Y}} f(x, y)$ as the primal function, and $y^*(x) = \arg\max_{\mathcal{Y}} f(x, y)$ as the optimal $y$ w.r.t $x$. Since the objective is nonconvex about $x$, we aim at finding an $\epsilon$-stationary point $(x_t, y_t)$ such that $\mathbb{E}\|\nabla_x f(x_t, y_t)\| \le \epsilon$ and $\mathbb{E}\|y_t - y^*(x_t)\| \le \epsilon$, where the expectation is taken over the randomness in the algorithm.

### 3.1 Convergence in deterministic and stochastic settings

**Assumption 3.3** (Stochastic gradients). *$\nabla_x F(x, y; \xi)$ and $\nabla_y F(x, y; \xi)$ are unbiased stochastic estimators of $\nabla_x f(x, y)$ and $\nabla_y f(x, y)$ and have variances bounded by $\sigma^2 \ge 0$.*

We assume the unbiased stochastic gradients have the variance $\sigma^2$, and the problem reduces to the deterministic setting when $\sigma = 0$. Now we provide a general analysis of the convergence for any adaptive optimizer used in the inner loop.

**Theorem 3.1.** *Define the expected cumulative suboptimality of inner loops as $\mathcal{E} = \mathbb{E}\left[\sum_{t=0}^{T-1} \frac{l^2\|y_t - y^*(x_t)\|^2}{2\sqrt{v_0}}\right]$. Under Assumptions 3.1, 3.2 and 3.3, the output from Algorithm 3 satisfies*

$$\mathbb{E}\left[\sqrt{\frac{1}{T}\sum_{t=0}^{T-1}\|\nabla_x f(x_t, y_t)\|^2}\right] \le \frac{2(A + \mathcal{E})}{\sqrt{T}} + \frac{v_0^{\frac{1}{4}}\sqrt{A + \mathcal{E}}}{\sqrt{T}} + \frac{2\sqrt{(A + \mathcal{E})\sigma}}{(MT)^{\frac{1}{4}}},$$

*where $A = \frac{2\Delta}{\eta} + \left(\frac{4\sigma}{\sqrt{M}} + 2\kappa l\eta\right)\left[1 + 2\log\left(\text{Poly}\left(T, \mathcal{E}, \frac{\Delta}{\eta}, \frac{\sigma}{\sqrt{M}}, \kappa l\eta, v_0, \frac{1}{v_0}\right)\right)\right]$.*

**Remark 2.** *The general analysis is built upon milder assumptions than existing work on AdaGrad in nonconvex optimization, not requiring either bounded gradient in [79] or prior knowledge about the smoothness modulus in [48]. This theorem implies the algorithm attains convergence for the nonconvex variable $x$ with any constant $\eta > 0$ and $v_0 > 0$ that does not depend on any problem parameter, so it is parameter-agnostic.*

**Remark 3.** *Another benefit of this analysis is that the variance $\sigma$ appears in the leading term $T^{-\frac{1}{4}}$, which means the convergence rate can interpolate between the deterministic and stochastic settings. It implies a complexity of $\widetilde{O}(\epsilon^{-2})$ in the deterministic setting and $\widetilde{O}(\epsilon^{-4})$ in the stochastic setting for the primal variable as long as the accumulated suboptimality for the inner-loops $\mathcal{E}$ is $\widetilde{\mathcal{O}}(1)$, regardless of the batch size $M$. However, $M$ can control the number of outer loops and there affect the sample complexity for the dual variable.*

In the next two theorems, we derive the total complexities, in the deterministic and stochastic settings, of finding $\epsilon$-stationary point by controlling the cumulative suboptimality $\mathcal{E}$ in Theorem 3.1 for subroutine $\mathcal{A}$ with specific convergence rate. In fact, we can also use any off-the-shelf adaptive optimizer for solving the inner maximization problem up to the desired accuracy. Note that (stochastic) GDmax fixes each inner-loop's accuracy or steps to be related with $\mu$, $\ell$ and $\epsilon$ so that $\mathcal{E}$ can be easily bounded [50, 64]. In contrast, since we do not have access to the problem parameters and $\epsilon$, Algorithm 3 gradually increases the inner-loop accuracy. In the proof of the following theorems, we will show that with our proposed stopping criteria and desired subroutines, $\mathcal{E}$ is bounded by $\mathcal{O}(\log T)$.

**Theorem 3.2** (deterministic). *Suppose we have a linearly-convergent subroutine $\mathcal{A}$ for maximizing any strongly concave function $h(\cdot)$:*

$$\|y^k - y^*\|^2 \le a_1(1 - a_2)^k\|y^0 - y^*\|^2$$

*where $y^k$ is $k$-th iterate, $y^*$ is the optimal solution, and $a_1 > 0$ and $0 < a_2 < 1$ are constants that can depend on the parameters of $h$. Under the same setting as Theorem 3.1 with $\sigma = 0$, for Algorithm 3 with $M = 1$ and a subroutine $\mathcal{A}$ under stopping criterion I, there exists $t^* \le \widetilde{O}\left(\epsilon^{-2}\right)$ such that $(x_{t^*}, y_{t^*})$ is an $\epsilon$-stationary point. Therefore, the total gradient complexity is $\widetilde{O}\left(\epsilon^{-2}\right)$.*

**Remark 4.** *This complexity is optimal in $\epsilon$ up to logarithmic term [86], similar to GDA [50]. Note that many adaptive and parameter-agnostic algorithms can achieve the linear rate when solving smooth and strongly concave maximization problems; to list a few, gradient ascent with backtracking line-search [75], SC-AdaNGD [44] and polyak stepsize [30, 54, 66] [4]. Here we can also pick more general subproblem accuracy in criterion I that only needs to scale with $1/t$.*

**Theorem 3.3** (stochastic). *Suppose we have a sub-linearly-convergent subroutine $\mathcal{A}$ for maximizing any strongly concave function $h(\cdot)$: after $K = k + 1$ iterations*

$$\mathbb{E}\|y^K - y^*\|^2 \leq \frac{b_1\|y^0 - y^*\|^2 + b_2}{k},$$

*where $y^k$ is $k$-th iterate, $y^*$ is the optimal solution, and $b_1, b_2 > 0$ are constants that can depend on the parameters of $h$. Under the same setting as Theorem 3.1 for Algorithm 3 with $M = \epsilon^{-2}$ and subroutine $\mathcal{A}$ under the stopping criterion II, there exists $t^* \leq \widetilde{O}\left(\epsilon^{-2}\right)$ such that $(x_{t^*}, y_{t^*})$ is an $\epsilon$-stationary point. Therefore, the total stochastic gradient complexity is $\widetilde{O}\left(\epsilon^{-4}\right)$.*

**Remark 5.** *This $\widetilde{O}\left(\epsilon^{-4}\right)$ complexity is nearly optimal in the dependence of $\epsilon$ for stochastic NC-SC problems [46]. Here we set $M = \epsilon^{-2}$ for the simplicity of exposition, and a similar result also holds for gradually increasing $M$. The sublinear rate specified above for solving the stochastic strongly convex subproblem can be achieved by several existing parameter-agnostic algorithms under some additional assumptions, such as FREEREXMOMENTUM [12] and Coin-Betting [13] [5]. Parameter-free SGD [9] is partially parameter-agnostic that only requires the stochastic gradient bound rather than the strongly-convexity parameter. Mukkamala and Hein [62] and Wang et al. [78] introduce the variants of AdaGrad, RMSProp and Adam for strongly-convex online learning, but they need to know both gradient bounds and strongly-convexity parameter for setting stepsizes. We will show in the next subsection that AdaGrad with a slower decaying rate is parameter-agnostic. We note that the analysis of this theorem is not the simple gluing of the outer loop and inner loop complexity, but requires more sophisticated control of the cumulative suboptimality $\mathcal{E}$.*

With the popularity of computational resource demanding deep neural networks, in both minimization and minimax applications, people find high probability guarantees for a single run of an algorithm useful [40, 49]. Given the lack of such guarantee in the minimax optimization, we provide a high probability convergence result for NeAda-AdaGrad in Appendix C, which shows a similar sample complexity as Theorem 3.3 under the subGaussian noise.

## 3.2 Generalized AdaGrad for strongly-convex subproblem

We now introduce the generalized AdaGrad for minimizing strongly convex objectives, which can serve as an adaptive subroutine for Algorithm 3, without requiring knowledge on the strongly convex parameter. We analyze it for the more general online convex optimization setting: at each round $t$, the learner updates its decision $x_t$, then it suffers a loss $f_t(x_t)$ and receives the sub-gradient of $f_t$. The generalized AdaGrad, described in Algorithm 4, keeps the cumulative gradient norm $v_t$ and takes the stepsize $\eta/v_t^\alpha$ with a decaying rate $\alpha \in (0, 1]$. When $\alpha = 1/2$, it reduces to the scalar version of AdaGrad [73]; when $\alpha = 1$, it reduced to the scalar version of SC-AdaGrad [62].

**Theorem 3.4.** *Consider Algorithm 4 for online convex optimization and assume that (i) $f_t$ is continuous and $\mu$-strongly convex, (ii) $\mathcal{X}$ is convex and compact with diameter $\mathcal{D}$; (iii) $\|g_t\| \leq G$ for every $t$. Then for $0 < \alpha < 1$ with any $\eta > 0$, the regret of Algorithm 4 satisfies:*

$$\max_{x \in \mathcal{X}} \sum_{t=0}^{T-1} (f_t(x_t) - f_t(x)) \leq c_\alpha + d_\alpha \left( v_0 + \sum_{t=1}^{T-1} \|g_t\|^2 \right)^{1-\alpha},$$

---

[4] Levy [44] needs to know the diameter of $\mathcal{Y}$. Hazan and Kakade [30], Loizou et al. [54], Orvieto et al. [66] use polyak stepsize which requires knowledge of the minimum or lower bound of the function value. AdaGrad achieves the linear rate if the learning rate is smaller than $O(1/l)$, and $O(1/k)$ rate otherwise [80].

[5] FREEREXMOMENTUM [12] and Coin-Betting [13] can achieves $\mathcal{O}(\log k/k)$ convergence rate when the stochastic gradient is bounded in $\mathcal{Y}$. If the subroutine has additional logarithmic dependence, it suffices to run the subroutine for $t \log^2(t)$ times using criterion II (see Appendix B).

---

**Algorithm 4** Generalized AdaGrad for Strongly-convex Online Learning

---

1: Input: $x_0$, $v_0 > 0$ and $0 < \alpha \le 1$ .
2: **for** $t = 0, 1, 2, ...$ **do**
3:     receive $g_t \in \partial f_t(x_t)$
4:     $v_{t+1} = v_t + \|g_t\|^2$
5:     $x_{t+1} = \mathcal{P}_\mathcal{X}\left(x_t - \frac{\eta}{v_{t+1}^\alpha}g_t\right)$
6: **end for**

---

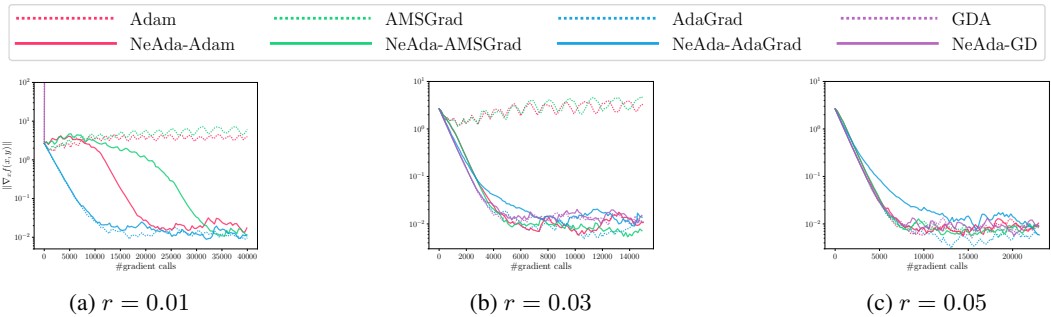

(a) $r = 0.01$          (b) $r = 0.03$          (c) $r = 0.05$

Figure 2: Comparison between the two families of non-nested and nested adaptive methods on McCromick function with stochastic gradient oracles. $\sigma = 0.01$, $\eta^y = 0.01$ and $r = \eta^y/\eta^x$.

*and for $\alpha = 1$ with $\eta \ge \frac{G^2}{2\mu}$,*

$$\max_{x \in \mathcal{X}} \sum_{t=0}^{T-1}(f_t(x_t) - f_t(x)) \le c_\alpha + d_\alpha \log\left(v_0 + \sum_{t=1}^{T-1}\|g_t\|^2\right),$$

*where $c_\alpha$ and $d_\alpha$ are constants depending on the problem parameters, $\alpha$ and $\eta$.*

The theorem implies a logarithmic regret for the case $\alpha = 1$, but the stepsize needs knowledge about problem's parameters $\mu$ and $G$; similar results are shown for SC-AdaGrad [62] and SAdam [78]. When $\alpha < 1$, the algorithm becomes parameter-agnostic and attains an $O(T^{1-\alpha})$ regret. Such parameter-agnostic phenomenon for smaller decaying rates is also observed for SGD in stochastic optimization [20]. Proving the regret bound for the generalized AdaGrad with $\alpha < 1$ in the online setting is challenging, since the adversarial $g_t$ can lead to a "sudden" change in the stepsize. In the proof, we bound the possible number of times such "sudden" change could happen.

To the best of our knowledge, this is the first regret bound for adaptive methods with general decaying rates in the strongly convex setting. By online-to-batch conversion [38], it can be converted to $O(T^{-\alpha})$ rate in the strongly convex stochastic optimization. Xie et al. [80] prove the $O(1/T)$ convergence rate, or a linear convergence rate when the smoothness parameter is known, for AdaGrad with $\alpha = 1/2$ in this setting, but under a strong assumption — Restricted Uniform Inequality of Gradients (RUIG) — that requires the loss function with respect to each sample $\xi$ to satisfy the error bound condition with some probability.

## 4 Experiments

To evaluate the performance of NeAda, we conducted experiments on simple test functions and a real-world application of distributional robustness optimization (DRO). In all cases, we compare NeAda with the non-nested adaptive methods using the same adaptive schemes. For notational simplicity, in all figure legends, we label the non-nested methods with the names of the adaptive mechanisms used. We observe from all our experiments that: 1) while non-nested adaptive methods can diverge without the proper two-time-scale, NeAda with adaptive subroutine always converges; 2) when the non-nested method converges, NeAda can achieve comparable or even better performance.

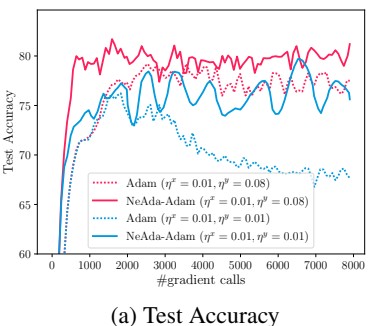

(a) Test Accuracy

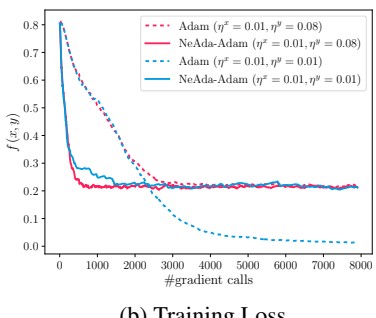

(b) Training Loss

Figure 3: Experimental results of distributional robustness optimization task on synthetic dataset.

## 4.1 Test functions

In Section 2, we have compared NeAda with non-nested methods on a quadratic function in Figure 1 and the observations match Lemma 2.1. Now we consider a more complicated function that is composed of McCormick function in $x$, a bilinear term, and a quadratic term in $y$,

$$f(x,y) = \sin(x_1 + x_2) + (x_1 - x_2)^2 - \frac{3}{2}x_1 + \frac{5}{2}x_2 + 1 + x_1y_1 + x_2y_2 - \frac{1}{2}(y_1^2 + y_2^2),$$

For this function, we compare the adaptive frameworks in the stochastic setting with Gaussian noise. As demonstrated in Figure 2, non-nested methods are sensitive to the selection of the two-time-scale. When the learning rate ratio is too small, e.g., $\eta^y/\eta^x = 0.01$, non-nested Adam, AMSGrad and GDA all fail to converge. We observe that GDA converges when the ratio reaches 0.03, while non-nested Adam and AMSGrad still diverge until 0.05. Although non-nested adaptive methods require a smaller ratio than GDA in Lemma 2.1, this example illustrates that adaptive algorithms sometimes can be more sensitive to the time separation. In comparison, NeAda with adaptive subroutine always converges regardless of the learning rate ratio.

## 4.2 Distributional robustness optimization

To justify the effectiveness of NeAda on real-world applications, we carried out experiments on distributionally robust optimization [71], where the primal variable is the model weights to be learned by minimizing the empirical loss while the dual variable is the adversarial perturbed inputs. The dual variable problem targets finding perturbations that maximize the empirical loss but not far away from the original inputs. Formally, for model weights $x$ and adversarial samples $y$, we have:

$$\min_x \max_{y=[y_1,\ldots,y_n]} f(x,y), \quad \text{where} \quad f(x,y) := \frac{1}{n}\sum_{i=1}^n f_i(x,y_i) - \gamma\|y_i - v_i\|^2,$$

where $n$ is the total number of training samples, $v_i$ is the $i$-th original input and $f_i$ is the loss function for the $i$-th sample. $\gamma$ is a trade-off parameter between the empirical loss and the magnitude of perturbations. When $\gamma$ is large enough, this problem is nonconvex-strongly-concave, and following the same setting as [71, 70], we set $\gamma = 1.3$. For NeAda, we use both stopping criterion I with stochastic gradient and criterion II in our experiments. For the results, we report the training loss and the test accuracy on adversarial samples generated from fast gradient sign method (FGSM) [26]. FGSM can be formulated as $x_{\text{adv}} = x + \epsilon \cdot \text{sign}(\nabla_x f(x))$, where $\epsilon$ is the noise level. To get reasonable test accuracy, NeAda with Adam as subroutine is compared with Adam with fixed 15 inner loop iterations, which is consistent with the choice of inner loop steps in [71], and such choice obtains much better test accuracy than the completely non-nested Adam. Our experiments include a synthetic dataset and MNIST [43] with code modified from [56].

**Results on Synthetic Dataset.** We use the same data generation process as in [71]. The inputs are 2-dimensional i.i.d. random Guassian vectors, i.e., $x_i \sim \mathcal{N}(0, I_2)$, where $I_2$ is the $2 \times 2$ identity matrix. The corresponding $y_i$ is defined as $y_i = \text{sign}(\|x_i\|_2 - \sqrt{2})$. Data points with norm in range

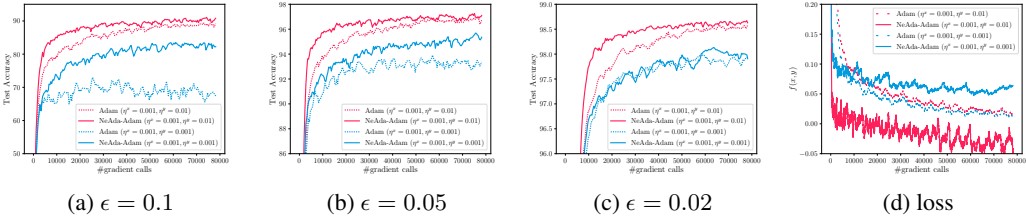

(a) $\epsilon = 0.1$          (b) $\epsilon = 0.05$          (c) $\epsilon = 0.02$          (d) loss

Figure 4: Results of distributional robustness optimization task on MNIST. $\epsilon$ is the noise level.

$(\sqrt{2}/1.3, 1.3\sqrt{2})$ are removed to make the classification margin wide. 10000 training and 4000 test data points are generated for our experiments. The model we use is a three-layer MLP with ELU activations.

As shown in Figure 3(a), when the learning rates are set to different scales, i.e., $\eta^x = 0.01, \eta^y = 0.08$ (red curves in the figure), both methods achieve reasonable test errors. In this case, NeAda has higher test accuracy and reaches such accuracy faster than Adam. If we change the learning rates to the same scale, i.e., $\eta^x = 0.01, \eta^y = 0.01$ (blue curves in the figure), NeAda retains good accuracy while Adam drops to an unsatisfactory performance. This demonstrates the adaptivity and less-sensitivity to learning rates of NeAda. In addition, Figure 3(b) illustrates the convergence speeds on the loss function, and NeAda (solid lines) always decreases the loss faster than Adam. Note that Adam with the same learning rates converges to a lower loss but suffers from overfitting, as shown in Figure 3(a) that its test accuracy is only about 68%.

**Results on MNIST Dataset.** For MNIST, we use a convolutional neural network with three convolutional layers and one final fully-connected layer. Following each convolutional layer, ELU activation and batch normalization are used.

We compare NeAda with Adam under three different noise levels and the accuracy is shown in Figures 4(a) to 4(c). Under all noise levels, NeAda outperforms Adam with the same learning rates. When we have proper time-scale separation (the red curves), both methods achieve good test accuracy, and NeAda achieves higher accuracy and converges faster. After we change to the same learning rates for the primal and dual variables (the blue curves), the accuracy drop of NeAda is slighter compared to Adam, especially when $\epsilon = 0.1$. As for the training loss shown in Figure 4(d), NeAda (the solid curves) is always faster at the beginning. We also observed that with proper time-scale separation, NeAda reaches a lower loss.

## 5 Conclusion

Both non-nested and nested adaptive methods are popular in nonconvex minimax problems, e.g., the training of GANs. In this paper, we demonstrate that non-nested algorithms may fail to converge when the time-scale separation is ignorant of the problem parameter even when the objective is strongly-concave in the dual variable with noiseless gradients information. We propose fixes to this problem with a family of nested algorithms–NeAda, that nests the max oracle of the dual variable under an inner loop stopping criterion. The proper stopping criterion will help to balance the outer loop progress and inner loop accuracy. NeAda-AdaGrad attains the near-optimal complexity without a priori knowledge of problem parameters in the nonconvex-strongly-concave setting. It can be a future direction to design parameter-agnostic algorithms for nonconvex-concave minimax problems or more general regimes by leveraging recent progress in nonconvex minimax optimization and the adaptive analysis in this paper. Another interesting direction is to investigate the convergence behavior of Adam-type algorithms with general decaying rates in the strongly convex online optimization.

## Acknowledgement

This work was supported by an ETH Research Grant funded through the ETH Zurich Foundation.

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
