# OpenReview forum: "Nest Your Adaptive Algorithm for Parameter-Agnostic Nonconvex Minimax Optimization"
_NeurIPS.cc/2022/Conference — NeurIPS 2022 Accept_

### Official Review · Reviewer_MNNk · 2022-07-03

**Rating:** 5
**Confidence:** 2
**Soundness:** 2 fair
**Presentation:** 3 good
**Contribution:** 2 fair

**Summary:**

This paper proposes an adaptive algorithm for non-convex-strongly-concave minimax optimization, where the step size does not depend on the problem parameters, e.g., $L$ of $L$-smoothness, and $\mu$ of $\mu$-strong concavity.
The algorithm has two loops: in the inner loop, the corresponding strongly-concave problem of $y$ given $x$ is approximately optimized by executing an off-the-shelf adaptive algorithm for a sufficiently number of iterations; while in the outer loop, an adaptive gradient descent step is used to update $x$.


**Questions:**

see above

**Strengths And Weaknesses:**

The topic of developing adaptive methods with theoretical guarantees is interesting. However, at the current point, I hesitate to recommend the acceptance for the following reasons, and I would like to get some clarifications.

1. While I understand the goal in this paper is to get an adaptive algorithm that does not need the knowledge of the problem parameters, getting a stationary-point guarantee on x for the class of non-convex-strongly-concave minimax optimization via applying adaptive methods on both x and y seems a bit straightforward to me, in my opinion. It would be more helpful for the evaluation if the challenge in the design and analysis of the algorithm can be highlighted.

2. Remark 2 for Theorem 3.1 says "the algorithm attains convergence for the nonconvex variable x with any choice of $\eta >0$".
But from line 185, if we set $\eta = T$, then $A=\Omega (T)$ and therefore the second term
$\frac{\sqrt{A+ ...} }{\sqrt{T}}$
on the r.h.s. of the bound of the expected gradient norm in Theorem 3.1 will be $\Omega(1)$. So the convergence does not hold for any step size $\eta$.


3. The purpose of introducing Algorithm 4 is not clear to me. Remark 5 mentions that FreeFREXMomenetum and Coin-Betting can be used for the inner loop of Algorithm 2. Given this, what is the point of having Algorithm 4?


4. (The first inequality on Page 22) How to get the last term on the r.h.s. of the first inequality on Page 22, i.e., $2 \kappa^2 \\| x_t - x_{t+1} \\|^2$?
It seems to me that the last term arises from bounding the last term on the last inequality on Page 21, i.e., by bounding $\\| y^*(x_t) -  y^*(x_{t+1}) \\|^2$. How to use Lemma A.1 to get the result?


Minor:

There seems a typo in the definition of Assumption 3.2, i.e., $\nabla_x f(x,y_2)$ <- $\nabla_y f(x,y_2)$.

---

> ### Author Response · Authors · 2022-08-02
> **To Reviewer MNNk (part 1/2)**
>
> Thanks for the detailed comments!
>
> >**Q1: While I understand the goal in this paper is to get an adaptive algorithm that does not need the knowledge of the problem parameters, getting a stationary-point guarantee on x for the class of non-convex-strongly-concave minimax optimization via applying adaptive methods on both x and y seems a bit straightforward to me. It would be more helpful for the evaluation if the challenge in the design and analysis of the algorithm can be highlighted.**
>
> We are afraid that the reviewer might have missed several crucial aspects of our contribution.
>
> First, we demonstrate in Section 2, both theoretically and numerically, that the non-nested adaptive methods in nonconvex minimax optimization do not always converge without knowledge about parameters. **Although existing literature has attempted this straightforward way of combining adaptive scheme on both $x$ and $y$, our simple example illustrates the impossibility for such method to be parameter-agnostic**. The main challenge lies in the necessity of problem-dependent two-time-scale separation for nonconvex minimax problems, which makes the design of adaptive methods fundamentally different from and more challenging than convex-concave regime or nonconvex minimization.
>
> Second, we introduce the nested adaptive framework that circumvents the challenge with problem-dependent time-scale separation by a two-loop structure with an adaptive stopping criterion from the inner loop. Our framework may seem straightforward in hindsight, but has not been properly studied in the literature before, at least to the best of our knowledge. Note that the adaptive stopping criterion and the generalized AdaGrad subroutine for strongly convex subproblems are rather new.
>
> Last and more importantly, our analysis of NeAda is quite nontrivial and yields several interesting theoretical contributions:
> - For the inner-loop analysis, our proof for NeAda-AdaGrad **requires different techniques from the existing analysis of GDmax** [1, 2]. As mentioned by Reviewer 2KvC, the accumulated error from the inner-loops is crucial for the complexity of nested algorithms , e.g., $\\mathcal{E} = \\mathbb{E}\\left[\\sum_{t=0}^{T-1} \\frac{l^{2}\\left\\|y_{t}-y^{\*}\\left(x_{t}\\right)\\right\\|^{2}}{2 \\sqrt{v_{0}}}\\right]$ in our paper. For non-adaptive algorithms such as GDmax, this term can be directly bonded by pre-fixing the inner-loop accuracy or steps to be dependant on $\\mu$, $l$ and $\\epsilon$. Since we do not have access these quantities, we have to increase accuracy gradually. Such practice requires us to: (a) bound the distance from the initial point of the inner-loop to its optimal point $y^*(x_t)$, (b) bound the sum of a non-trivial sequence of decreasing errors (we refer to the proofs of Theorem 3.2 and Lemma A.3, or Lemma A.4 in the original version).
>
> - For the outer-loop analysis, we also came up a new proof that removes the restrictive bounded gradient assumption that is commonly used in AdaGrad-type methods, as shown in our general result in in Theorem 3.1.
>
> - Another result of independent interest is our novel analysis of parameter-free AdaGrad methods for strongly convex losses, which seems absent in the literature.
>
> >**Q2: The convergence does not hold for any step size $\\eta$.**
>
> Here we mean the convergence rate holds for any $\\eta = \\Theta(1)$ that is independent from problem parameters, $T$ or $\\epsilon$. Since $\\eta$ does not need to depend on problem parameters, such as $\\mu$ and $l$, we call it problem-agnostic. In the updated version, we made this point more clear.
>
>
> *continued below*
>
>
> **References**
>
> [1] T. Lin, C. Jin, and M. Jordan. On gradient descent ascent for nonconvex-concave minimax problems. In *ICML*, pages 6083–6093. PMLR, 2020.
>
> [2] M. Nouiehed, M. Sanjabi, T. Huang, J. D. Lee, and M. Razaviyayn. Solving a class of non- convex min-max games using iterative first order methods. *NeurIPS*, 32, 2019.

---

> > ### Author Response · Authors · 2022-08-02
> > **To Reviewer MNNk (part 2/2)**
> >
> > >**Q3: what is the point of having Algorithm 4?**
> >
> > Although NeAda allows different adaptive schemes for the inner-loop, choosing **AdaGrad for maximizing $y$ is still practical and widely used**. Our analysis provides a convergence guarantee for this, and explains why using AdaGrad for both sides leads to a good performance in the experiments. We strongly believe **this result is also an important contribution in online learning with adaptive methods**, which is also pointed out by Reviewer 2KvC. To the best of our knowledge, it is the first time that the parameter-agnostic property is noticed for minimizing strongly-convex objective with the adaptive method and a general decaying rate $\\alpha<1$. In contrast, previous literature [3, 4] only considers the case $\\alpha=1$ which requires knowledge about problem parameters. Providing the regret bound with $\\alpha<1$ is much more technically challenging as we can see in the proof of Theorem 3.4. In the updated version, we come up with a simpler proof with the similar idea as our previous proof, and the new proof gets rid of the logarithmic term and leads to a data-dependent regret bound for $\\alpha<1$.
> >
> > >**Q4: How to use Lemma A.1 to bound $\\left\\|y^{\*}\\left(x_{t}\\right)-y^{\*}\\left(x_{t+1}\\right)\\right\\|^{2}$**
> >
> > Thanks for pointing this out! According to Lemma 4.3 in [5] or Lemma A.3 in [6], $y^\*(\\cdot)$ is $\\kappa$-Lipschitz where $y^\*(x) := \\mathrm{argmax}_{y \\in \\mathcal{Y}} f(x,y)$. We have added this part to Lemma A.1 in our updated version.
> >
> > We hope our answers address the reviewer’s confusions and kindly ask the reviewer to re-evaluate our work based on the above clarifications.
> >
> > **References**
> >
> > [3] M. C. Mukkamala and M. Hein. Variants of rmsprop and adagrad with logarithmic regret bounds. In *ICML*, pages 2545–2553. PMLR, 2017.
> >
> > [4] G. Wang, S. Lu, Q. Cheng, W.-w. Tu, and L. Zhang. Sadam: A variant of adam for strongly convex functions. In *ICLR*, 2020.
> >
> > [5] T. Lin, C. Jin, and M. Jordan. On gradient descent ascent for nonconvex-concave minimax problems. In *ICML*, pages 6083–6093. PMLR, 2020.
> >
> > [6] M. Nouiehed, M. Sanjabi, T. Huang, J. D. Lee, and M. Razaviyayn. Solving a class of non- convex min-max games using iterative first order methods. *NeurIPS*, 32, 2019.

---

> > ### Comment · Reviewer_MNNk · 2022-08-09
> > **comment**
> >
> > I just have a quick question regarding Q2.
> >
> > In the literature, do those related works of adaptive algorithms not allow \eta = T? If most of the works of adaptive algorithms do not allow \eta = T, then I am fine with the authors' response. To answer my question, the authors might want to give some examples, e.g., Theorem X in [ABC].
> >
> >
> > I will read the authors' reply and other comments later, and update the score if it fits. Thanks.

---

> > > ### Author Response · Authors · 2022-08-09
> > > **Response to the question about \eta = T**
> > >
> > > Thanks for the discussion.
> > >
> > > None of the existing analysis of adaptive methods could allow such a large stepsize as $\\eta=O(T)$. For AdaGrad, see e.g., Theorem 7 in [1] and Theorem 2.1 in [2], which require stepsize $\\eta = O(1)$ to reach the optimal rate; for AMSGrad, see e.g., Theorem 4 in [3], which requires even smaller stepsize $\\eta = O(1/\\sqrt{T})$. Moreover, as the reviewer asked, from all those theorems, setting $\eta =O(T)$ would naturally lead to diverge.
> > >
> > > References:
> > >
> > >
> > > [1] John Duchi, Elad Hazan, and Yoram Singer. Adaptive subgradient methods for online learning and stochastic optimization. *Journal of machine learning research*, 12(7), 2011.
> > >
> > > [2] Rachel Ward, Xiaoxia Wu, and Leon Bottou. Adagrad stepsizes: Sharp convergence over nonconvex landscapes. In *ICML*, pages 6677–6686. PMLR, 2019.
> > >
> > > [3] Sashank J Reddi, Satyen Kale, and Sanjiv Kumar. On the convergence of adam and beyond. In *ICLR*, 2018.

---

> > > > ### Comment · Reviewer_MNNk · 2022-08-09
> > > > **comment**
> > > >
> > > >
> > > > I found it quite concerning that there is a component,
> > > > $$\frac{ \Phi(x_0) - \min_x \Phi(x) }{ \eta } + \kappa l \eta$$
> > > > in the bound of the convergence rate (see Theorem 3.1) of the proposed adaptive algorithm.
> > > >
> > > > This component is similar to the regret bound of the standard (non-adaptive) online gradient descent. My concern is that a regret bound of a non-adaptive method appears in the guarantee of proposed adaptive method does not seem natural. Other related works of adaptive methods do not seem to have such a component (but correct me if I am mistaken), see
> > > > e.g.,
> > > >
> > > > [Kavis et al.] UniXGrad: A Universal, Adaptive Algorithm with Optimal Guarantees for Constrained Optimization.
> > > >
> > > > [Levy et al. ] Online Adaptive Methods, Universality and Acceleration
> > > >
> > > > If one set $\eta = 20$, then Theorem 3.1 just means that the gradient norm is $\Omega(1)$ for the first $T=20$ iterations, which doesn't seem a natural feature of the adaptive algorithms in the literature.  Also, I believe "None of the existing analysis of adaptive methods" is a bit misleading, because [Kavis et al.] and [Levy et al.] do not seem to have a natural counterpart of $\eta$ as this work for the user to specify. Their works do not have the issue that I am raising here.
> > > >
> > > > I hope the other reviewers and the area chair will help check this issue.

---

> > > > > ### Author Response · Authors · 2022-08-09
> > > > > **Response to the question about \eta**
> > > > >
> > > > > **Other related works of adaptive methods do not seem to have such a component.**
> > > > >
> > > > > Two works mentioned in our last response have similar components (in the theorems we cited).
> > > > >
> > > > > * Theorem 7 in [1]: $\\left(\\frac{1}{\\textcolor{red}{\\eta}} \\\|x^*\\\|^2_2 +  \\textcolor{red}{\\eta} \\right) \mathrm{tr}(G\_T^{1/2})$.
> > > > > * Theorem 2.1 in [2]: $\\mathcal{Q}=\\frac{F\\left(x\_{0}\\right)-F^{*}}{\\textcolor{red}{\\eta}}+\\frac{4 \\sigma+\\textcolor{red}{\\eta} L}{2} \\log \\left(\\frac{20 N\\left(\\gamma^{2}+\\sigma^{2}\\right)}{b\_{0}^{2}}+10\\right)$.
> > > > >
> > > > > All these works and our paper, we can simply pick $\eta = 1$ or other constants, which can make the term $\frac{\Phi\left(x_{0}\right)-\min _{x} \Phi(x)}{\eta}+\kappa l \eta = O(1)$.
> > > > >
> > > > > **Works listed by the reviewer do not have such a component.**
> > > > >
> > > > > The two works the reviewer mentioned do not have $\\eta$ because they already set $\eta$ to be a specific constant in the very beginning.
> > > > > Comparing to line 5 of our Algorithm 3,
> > > > >
> > > > > * In [Kavis et al.], they just set $\\eta = 2D$ according to their equation 7, where $D$ is the diameter of the feasible set.
> > > > > * In [Levy et al.], they just set $\\eta = D/\\sqrt{2}$ according to their Algorithm 1.
> > > > >
> > > > > **More explanations to the reviewer's confusion.**
> > > > >
> > > > > We are afraid that the reviewer might have overlooked key notation differences, which may have led to the confusion:
> > > > >
> > > > > 1. Learning rates: in non-adaptive method,  $\\eta\_t = \\eta$; in our adaptive method, $\\eta\_t= \\eta/\\sqrt{v\_{t+1}}$. In our case, $\\eta$ is a hyperparameter but not the "true stepsize" $\\eta\_t$ (or we called effective stepsize in our paper).
> > > > >
> > > > > 2. Regret bound: typical bound for non-adaptive method on gradient norm is of form $\\textcolor{red}{O(\sqrt{\\frac{\\Delta}{\\eta T} + \\eta })}$, where the optimal choice for $\eta = \Theta( 1/\sqrt{T})$ and needs knowledge about problem parameters; our bound is of form $\\textcolor{red}{O((\\frac{\\Delta}{\\eta} + \\eta)/T^{1/4})}$, where the optimal choice for $\eta$ is a constant.
> > > > > In the non-adaptive case, the choice of $\\eta$ needs to depend on $T$ and other parameters; but in the adaptive case, the choice of $\\eta$ (as hyperparameter) can be any constant, which exactly showcases the parameter-agnostic feature of adaptive methods.
> > > > >
> > > > >
> > > > > References:
> > > > >
> > > > >
> > > > > [1] John Duchi, Elad Hazan, and Yoram Singer. Adaptive subgradient methods for online learning and stochastic optimization. *Journal of machine learning research*, 12(7), 2011.
> > > > >
> > > > > [2] Rachel Ward, Xiaoxia Wu, and Leon Bottou. Adagrad stepsizes: Sharp convergence over nonconvex landscapes. In *ICML*, pages 6677–6686. PMLR, 2019.
> > > > >
> > > > >
> > > > > [Kavis et al.] UniXGrad: A Universal, Adaptive Algorithm with Optimal Guarantees for Constrained Optimization.
> > > > >
> > > > > [Levy et al.] Online Adaptive Methods, Universality and Acceleration.

---

> > > > > > ### Comment · Reviewer_MNNk · 2022-08-09
> > > > > > **Thanks for the detailed answer**
> > > > > >
> > > > > > I upgrade the score accordingly.

---

> ### Author Response · Authors · 2022-08-08
> **Author-Reviewer discussion period**
>
> Dear Reviewer MNNk,
>
> As the discussion period is ending soon today, we appreciate if the reviewer could at least take a chance to read our responses. Have we addressed your main questions regarding the motivation of the adaptive algorithms and challenges behind? Do you have any further questions that we can clarify?

---

### Official Review · Reviewer_VNUE · 2022-07-22

**Rating:** 7
**Confidence:** 2
**Soundness:** 3 good
**Presentation:** 3 good
**Contribution:** 3 good

**Summary:**

Contribution summary:
The authors investigate two frameworks for adaptive minimax optimization: one is a simple (non-nested) adaptive framework, which performs one step of update of x and y simultaneously with adaptive gradients; the other is the nested adaptive (NeAda) framework and
provide the convergence analysis for a representative of NeAda that uses AdaGrad step size. In addition, there are two other complementary contributions which are a proposed AdaGrad step size and high probability convergence results. The authors also include empirical results for comparison with existing algorithms.

**Questions:**

N/A

**Limitations:**

See weaknesses.

**Strengths And Weaknesses:**

Strengths:
- First adaptive nonconvex minimax optimization algorithm? There exists literature for adaptive minimax optimization but not for nonconvex.
- The existing NCSC optimization algorithm has O(1/eps^2) oracle complexity and O(1/eps^4) sampling complexity. While the algorithm achieved the same rate, it does not require knowledge of the parameters like existing algorithms. The analysis of the algorithms does not follow trivially from existing literature. Some results such as theorem 3.1 also do not require common assumptions such as bounded gradient.
- Showed an example where GDA/non-nested methods does not work without prior knowledge of the parameters.


Weaknesses:
It incorporates AdaGrad step size which is widely known, but it seems the theoretical contribution is significant enough.

The algorithm does not necessary converge faster existing algorithms but it does have the advantage of being robust.

---

> ### Author Response · Authors · 2022-08-02
> **To Reviewer VNUE**
>
> Thanks for recognizing the contributions of our results! We made a few revisions as pointed by our general comments to further improve our paper.

---

### Official Review · Reviewer_2KvC · 2022-07-22

**Rating:** 6
**Confidence:** 3
**Soundness:** 3 good
**Presentation:** 3 good
**Contribution:** 2 fair

**Summary:**

The paper proposes an adaptive framework, called NeAda, for solving nonconvex-strongly concave (NC-SC) minimax problems which achieves the best-known rates both in the deterministic and stochastic settings up to logarithmic factors, while being agnostic to the problem parameters. These results are achieved when the NeAda framework is run with AdaGrad for the minimizing player. They also provide a set of independent but complementary results such as non-convergence of gradient descent ascent, the high probability convergence of NeAda-AdaGrad and regret bounds for a generalized parameter-agnostic AdaGrad template. The authors also provide a set of experiments demonstrating advantages of their framework.

**Questions:**

__Small comments:__
- Theorem 3.1 is referred to as Lemma 3.1 in line 199. It is referred to as a "lemma" in multiple places in the paper.
- What is $\xi_t$ in Algorithm 4, line 4?

__Questions:__
- Lemma 2.1 provides divergence result in terms of the last iterates. I wonder whether there would be any conditions on the learning rate ratios that implies divergence when we consider the average iterates.
- As far as I understand, Ne-Ada framework proposes a concrete parameter-agnostic subroutine for the _minimizing_ player with adaptive step-sizes, however, it assumes the existence of a parameter-agnostic linearly-convergent algorithm $\mathcal A$ without proposing a proof for a particular algorithm $\mathcal A$. Is there any linearly-convergent parameter-agnostic algorithm that works under the precise set of assumptions that the authors propose in the Theorem statement 3.1?
- Although the analysis of NeAda requires a linearly-convergent algorithm to achieve the almost-optimal rates, the regret rate in Theorem 3.4 does not imply that Algorithm 4 converges linearly. Hence it may not satisfy the theorem statement. Is there an immediate extension of the regret rate in Theorem 3.4 that would yield a linear rate in the offline setting? To my knowledge there is no such conversion scheme.


**Limitations:**

One comments I have is with respect to subroutine $\mathcal A$ in Algorithm 3. Authors talk about some candidates but there is no clear discussion of whether any such algorithm satisfies the set of assumptions of Theorem 3 while being parameter-agnostic. I would suggest to have a more thorough discussion on this. Other than that, I think authors discuss the limitations and shortcomings of their results with respect to the literature.

To my knowledge, this work does not seem to have any potential negative societal impact.

**Strengths And Weaknesses:**

__Note__: _I was assigned to the paper as an emergency reviewer, therefore I haven’t had long enough time to have a detailed look at the paper and the supplementary. My score is given with caution based on the points that I am confused about. I am willing to engage in discussions with the authors and possibly update my score accordingly. In case I made mistakes or misinterpreted anything while evaluating the work, please indicate them for me so that I will provide a better review and fair judgment of the paper._

The paper is structured in a good way; listing the contributions early on and building upon the idea of parameter-freeness. The possible empirical advantages of the nested framework that runs for multiple ascent iteration per update of the minimizing player are also presented to have a nice basis for the theoretical claims.

The proposed setting with the set of assumptions relaxes some of the restrictive assumptions in the literature, one of which is the removal of bounded stochastic gradients assumption. This assumption appears frequently in the analysis of adaptive AdaGrad-type methods and its removal is a good step forward in my opinion.

Lemma 2.1 provides a set of nice but fairly expectable results (with analogous results known for convex-concave setting). With Figure 1, I think it complements one of the main arguments around the learning rate ratios and shows how NeAda framework is less sensitive to ill-approximation of such a ratio.

Presentation of Theorems 3.1, 3.2 and 3.3 are accompanied with relevant remarks and clarifications. The convergence rate is noise-adaptive, which I think is important.

As an independent results, Theorem 3.4 is a nice contribution in terms of parameter-free online learning and adaptive regret bounds for strongly convex losses.

Experimental results are somewhat encouraging; Figure 2 demonstrates possible scenarios where the nested algorithm proves useful, while Figure 3 gives some clues about the performance improvements it would offer for optimization/generalization tasks.

--

From the very beginning, it must be clarified for which adaptive algorithms the paper provides the convergence analysis and rates under the nested structure. The first bullet point at line 56 suggests that NeAda framework provably works with multiple adaptive algorithms, whereas the paper focuses on AdaGrad and its theoretical behavior. Although the second bullet point clearly states that the almost-optimal rates are achieved for NeAda-AdaGrad, I would suggest to make it clear at all parts of the paper that theoretical results are presented for NeAda-AdaGrad only.

The statement of Theorem 3.2 is not precise enough; it is not clear what quantities are bounded and how it implies the total sample complexity (e.g., no clear reference to $\mathcal E$). I would suggest the same for Theorem 3.3. I believe using the versions from the appendix, or some modification of theirs, would be better.

The quantity $\mathcal E$ is actually a crucial part of the total sample complexity, and I would suggest to talk about it before Theorem 3.1. This would give some grounds for the results in Theorem 3.2 and 3.3 while explaining implications of the convergence rate in Theorem 3.1

One important contribution of this paper is enabling any linearly-convergent algorithms for strongly convex-minimization to be used in the NeAda-AdaGrad algorithm. However, this also creates a problem. From my understanding, parameter-agnostic property depends on the subroutine $\mathcal A$ and its properties. As long as there exists a parameter-free algorithm that converges linearly under the assumptions of Theorem 3.1 for strongly convex minimization problems, the claims hold. However, I am not sure whether such an algorithm exists. Could the authors please clarify this point for me?

---

> ### Author Response · Authors · 2022-08-02
> **To Reviewer 2KvC (part 1/2)**
>
> We thank the reviewer for the comprehensive review and valuable comments despite the short time as an emergency reviewer. Below we address the questions brought up by the review and are open to engage in more discussions with the reviewer if there are further questions.
>
> >**Weakness 1: From the very beginning, it must be clarified for which adaptive algorithms the paper provides the convergence analysis and rates under the nested structure. ... I would suggest to make it clear at all parts of the paper that theoretical results are presented for NeAda-AdaGrad only.**
>
> Thanks for the comment. In the revision, we re-emphasized that our convergence complexity is provided for NeAda-AdaGrad in our abstract and several other places. We would also like to remark that Section 2, which corresponds to first bullet of our contribution, is a divergence result for general adaptive schemes, including AdaGrad, Adam, AMSGrad, etc. We show both analytically and empirically that these non-nested methods will not converge. We believe that our convergence analysis for NeAda-AdaGrad in Section 3 also sheds light on other adaptive schemes and validates the robustness of using a nested framework.
>
> >**Weakness 2: For Theorem 3.2 and 3.3, using the versions from the appendix, or some modification of theirs, would be better. The quantity $\\mathcal{E}$ is actually a crucial part of the total sample complexity, and I would suggest to talk about it.**
>
> Thanks for the suggestions! We have incorporated the comments in our revision. We replaced Theorem 3.2 and 3.3 by the more detailed versions, and we have more discussion about $\\mathcal{E}$ before Theorem 3.2. Note the term $\\mathcal{E}$ depends on the proposed stopping criterion and subroutine used in the inner loop. In the deterministic setting, our stopping criterion readily guarantees that $\\mathcal{E} = O(\\log T)$. In the stochastic setting, the stopping criterion and a subroutine with $\\tilde{O}(1/k)$ rate will also guarantees $\\mathcal{E} = O(\\log T)$, as shown in the proof of Theorem 3.3. In Theorem 3.2 and 3.3, the algorithm will find an $\\epsilon$-stationary point, which is a point $(x, y)$ such that $\\mathbb{E}\\|\\nabla_x f(x, y)\\| \\leq \\epsilon$ and $\\mathbb{E}\\|y - y^\*(x) \\| \\leq \\epsilon$ as defined in our main paper.
>
> >**Weakness 3: One important contribution of this paper is enabling any linearly-convergent algorithms for strongly convex-minimization to be used in the NeAda-AdaGrad algorithm..... However, I am not sure whether such an algorithm exists. Could the authors please clarify this point for me?**
>
> We are afraid there might be a misunderstanding here. We would like to clarify that to achieve the near-optimal rate for NeAda-AdaGrad, **it only requires linearly-convergent subroutine for deterministic setting and sub-linearly-convergent subroutine with $\\boldsymbol{\\tilde{O}(1/k)}$ rate for stochastic setting**. As explained in Theorem 3.3 or Theorem B.2, we only require a subroutine that achieves $O(1/k)$ accuracy after $\\tilde{\\Theta}(k)$ steps for the stochastic setting. We realized that the confusion might be caused by a small typo in the statement of Theorem B.2 in appendix for the stochastic case: it should be “sub-linearly” rather than “linearly”, which we have now fixed.
>
> These rates of subroutines are reasonable for the strongly-concave objectives and can be achieved by many existing algorithms. In the deterministic setting, the linear convergence rate can be achieved by parameter-agnostic algorithms such as GD with line-search, Polyak stepsize and other algorithms in Remark 4. In the stochastic setting, the $\\tilde{O}(1/k)$ convergence rate can be achieved by FREEREXMOMENTUM and Coin-Betting as mentioned in Remark 5.
>
> *continued below*

---

> > ### Author Response · Authors · 2022-08-02
> > **To Reviewer 2KvC (part 2/2)**
> >
> > >**Q1: Lemma 2.1 provides divergence result in terms of the last iterates. I wonder whether there would be any conditions on the learning rate ratios that implies divergence when we consider the average iterates?**
> >
> > Thanks for the great question! The averaged iterate or randomly picked iterate are often used in convex and non-convex optimization. Yes, the **same condition** for the example in Lemma 2.1 will lead to divergence of non-nested methods even for average iterates. This can be derived from two observations: (a) The set of stationary point is the line $y = Lx$. When an iterate $(x_t, y_t)$ is on one side of the line, the next iterate $(x_{t+1}, y_{t+1})$ will stay on the same side of the line by the update rule and the gradients of this example; (b) The distance of the iterate $(x, y)$ to the line $y = Lx$ is $\\frac{|Lx- y|}{\\sqrt{L^2 + 1}}$ , which is proportional to the norm of the gradient about $x$. So the proof Lemma 2.1 directly implies that the distance from the iterate to the stationary point set is increasing with $t$. Therefore, either averaged iterate or randomly picked point from the past iterates will not converge. Detailed proof is provided in the appendix of our revision.
> >
> > >**Q2: Is there any linearly-convergent parameter-agnostic algorithm that works under the precise set of assumptions that the authors propose in the Theorem statement 3.1?**
> >
> > Please find our answers above. For the stochastic setting, our framework only requires a sub-linearly convergent subroutine with $\\tilde{O}(1/k)$ rate. Such convergence rate can be achieved by parameter-agnostic algorithms such as FREEREXMOMENTUM and Coin-Betting mentioned in Remark 5.
> >
> > >**Q3: The regret rate in Theorem 3.4 does not imply that Algorithm 4 converges linearly.**
> >
> > We provide the regret bound in Theorem 3.4 mainly for the stochastic setting. It implies $\\tilde{O}(1/k^{\\alpha})$ convergence rate for maximizing a strongly-concave function with $\\alpha \\leq 1$. We first notice the parameter-agnostic property of generalized AdaGrad for $\\alpha <1$ and fill the theoretical gap for adaptive methods with general decaying rates. In the updated version, we propose a simpler proof for Theorem 3.4 under the similar idea as the previous one. The new proof leads to a better data-dependant regret bound for $\\alpha < 1$, which depends on the past $\\{g_s\\}_{s \\leq k}$ and is bounded by $O(k^{1-\\alpha})$ in the worst case. We also note that a modified AdaGrad, SC-AdaNGD [1], will be able to achieve the linear rate in the deterministic setting.
> >
> > >**Q4: Theorem 3.1 is referred to as Lemma 3.1 in line 199. What is $\\xi_t$ in Algorithm 4, line 4?**
> >
> > Thanks for pointing out the typos! We have fixed them. Line 4 should simply be $v_{t+1}=v_{t}+\\|g_t\\|^2$.
> >
> > **References**
> >
> > [1] K. Levy. Online to offline conversions, universality and adaptive minibatch sizes. *NeurIPS*, 30, 2017.

---

> ### Author Response · Authors · 2022-08-08
> **Author-Reviewer discussion period**
>
> Dear Reviewer 2KvC,
>
> As the discussion period is ending soon (today), we appreciate if the reviewer could at least take a chance to read our responses. Have we addressed your main questions regarding the linear-convergent subroutine? Do you have any further questions that we can clarify?

---

> > ### Comment · Reviewer_2KvC · 2022-08-09
> > **Thank you for your clarifications**
> >
> > I have gone through the responses and clarifications provided by the authors. I believe some of the major concerns were addressed in the response, hence I am increasing my score.

---

> > > ### Author Response · Authors · 2022-08-09
> > > **Thank you for reading our responses**
> > >
> > > We thank the reviewer for acknowledging our responses and we are open to discussion if more questions come up.

---

### Author Response · Authors · 2022-08-02
**To all reviewers**

Thanks for the suggestions and we have updated the manuscript accordingly. Here are some updates in the new revision:
- We replaced Theorem 3.2 and 3.3 by a modified version of Theorem B.2 and B.3 in the appendix. As suggested by Reviewer 2KvC, the detailed statement in the original appendix is easier for readers to understand.
- We made it clear that our main convergence results are for NeAda-AdaGrad, as suggested by Reviewer 2KvC. We changed the claims “NeAda can achieve ... complexity” to “NeAda with AdaGrad stepsizes” or “NeAda-AdaGrad” in the abstract and other places.
- We added more discussion on the term $\mathcal{E}$, which is the accumulated suboptimality in the inner-loops, before the statement of Theorem 3.2. As pointed out by Reviewer 2KvC, this term is crucial for deriving the complexity.
- We added in Remark 1 that Lemma 2.1 also applies to the averaged or best iterate, according to a question from Reviewer 2KvC. We then provided the proof in the appendix.
- We simplified the proof of Theorem 3.4 with a new technical Lemma A.2. The new proof shares similar idea as before but brings two additional benefits: (i) it gets rid of the logarithmic term in the case $\alpha$ < 1; (ii) it makes the regret bound data-dependant in the case $\alpha$ < 1.
- According to a question raised by Reviewer MNNk, we changed the claim “the algorithm attains convergence for the nonconvex variable x with any choice of $\eta$ > 0” in Remark 2 to be “any constant $\eta$ > 0”.
- Lemma A.1 now includes the Lipschitzness of $y^*(\cdot)$, as pointed out by Reviewer MNNk.

---

### Meta-Review · Area_Chair_DfW6 · 2022-08-28

**Recommendation:** Accept
**Confidence:** Certain

**Metareview:**

During the rebuttal it was agreed that this paper provides good explanations regarding the machineries
that allow to guarantee convergence for minimax non-convex - strongly-concave problems.
The nesting technique together with the adagrad stepsize are nicely composed, and the proof technique is non-trivial. I therefore recommend to accept.

**Award:**

No

---

### Decision · Program_Chairs · 2022-09-14

Accept